# Impact of preschool attendance, parental stress, and parental mental health on internalizing and externalizing problems during COVID-19 lockdown measures in preschool children

**Irina Jarvers***, **Angelika Ecker, Daniel Schleicher, Romuald Brunner, Stephanie Kandsperger**

Department of Child and Adolescent Psychiatry and Psychotherapy, University of Regensburg, Regensburg, Germany

* irina.jarvers@ukr.de

**Data Availability Statement:** All data files including a variable description are available from the OSF database (Link: https://osf.io/2c4dk/).

## Abstract

### Background

Internalizing problems are common in young children, often persist into adulthood, and increase the likelihood for subsequent psychiatric disorders. Problematic attachment, parental mental health problems, and stress are risk factors for the development of internalizing problems. COVID-19 lockdown measures have resulted in additional parental burden and especially their impact on preschool children has rarely been investigated as of now. The current study examined the impact of sustained preschool attendance, parental stress, and parental mental health on internalizing and externalizing problems during COVID-19 lockdown measures in a sample of preschool children in Germany.

### Methods and findings

$N$ = 128 parents of preschool children filled out a one-time online survey about children's internalizing problems, externalizing problems, and attachment for three time points: before a nation-wide lockdown (T1), during the most difficult time of the lockdown (T2) and after the lockdown (T3). Additionally, parents answered questions about their own depressive and anxious symptomatology for the three time points and parental stress for T1 and T2. Linear-mixed effect models were computed to predict children's internalizing / externalizing behavior. Preschool children showed a significant increase in internalizing and externalizing problems over time, highest at T2 with small decreases at T3. Parental depressive and anxious symptomatology increased significantly from T1 to T2, but also remained high at T3. Parental stress levels were comparable to community samples at T1, but attained average values reported for at-risk families at T2. Linear-mixed effect models identified higher parental stress, parental anxiety, attachment problems, parental education, and less preschool attendance as significant predictors for internalizing and externalizing problems in

**Funding:** The author(s) received no specific funding for this work.

**Competing interests:** The authors have declared that no competing interests exist.

preschoolers with more specific associations shown in separate models. A limitation is the retrospective assessment for the times T1 and T2.

## Conclusions

Preschool children's mental health is strongly and negatively influenced by the ongoing COVID-19 pandemic and its lockdown measures. Sustained preschool attendance may serve as a protective factor.

## 1. Introduction

It is estimated that up to 15 % of children and adolescents worldwide suffer from mental health problems [1], which constitute the number one cause of disability [2]. In consequence of the ongoing COVID-19 pandemic, even larger percentages have been reported [3–6]. Most commonly, mental health problems are split into externalizing problems (i.e., aggression, hyperactivity and oppositional defiance) and internalizing problems (i.e., anxiety, depression, social withdrawal and somatic complaints) [7]. Externalizing problems constitute behaviors directed towards the environment and decrease with age, whereas internalizing problems constitute behaviors directed towards the self and increase with age [8]. Examining externalizing and internalizing problems early on is relevant, as they are predictive of later mental health disorders in adolescence [9], as well as adulthood [10, 11]. Although both types of problems co-occur in young children [12], internalizing problems are more likely to be overlooked by parents and educators due to their orientation towards the self [13–15].

Internalizing problems are present in up to 20 % of children and have demonstrated continuity from early to mid-childhood [16, 17]. Risk factors for the presence of internalizing problems can on the one hand apply to children themselves such as child temperament [18, 19]. On the other hand, environmental factors such as parenting style [20], parental mental health [21–23], parental stress [24, 25], peer experiences [26], and socio-economic status (SES; [27] have been identified as predictors for internalizing problems. However, most studies examining internalizing problems were conducted with adolescents or primary school children. A focus on preschool children is of particular importance as mental health problems require early interventions and preschool children struggle to verbalize their difficulties [28].

In most studies with preschool children, parental stress and parental mental health are the two environmental factors that have demonstrated the strongest relationship with children's internalizing problems. In a study by Sher-Censor et al. [29], maternal stress was strongly associated with toddlers' internalizing problems. In terms of parental mental health, maternal depression was most commonly examined and showed a strong positive relationship in a longitudinal study predicting internalizing problems at 5 years of age [30]. In another longitudinal study over 15 years, Côté et al. [31] showed that exposure to maternal depression early on (before the age of 5) was a predictor for internalizing disorders such as depression, anxiety and social phobia.

In addition to parental factors, also the attachment between child and parent impacts mental health in preschool children, as insecure attachment was shown to related to internalizing problems in early childhood [32, 33]. Overall, this suggests that during early childhood, attachment, but also parental stress and parental mental health appear to impact the development of internalizing problems.

Under some circumstances, risk factors are more likely to emerge and the ongoing global COVID-19 pandemic is one such case [34]. In order to control the spread of the COVID-19

virus, nationwide lockdowns were used across the globe, involving social distancing measures, closures of restaurants and shops and in particular school and preschool closures. Lockdown measures have been shown to result in increased parental burnout [35], parental stress [36] and clinically significant parental depressive and anxious symptomatology [37]. In this context, parents of younger children were more affected than parents of older children [38]. Some of the reasons mentioned are especially closures of schools and preschools, as well as social distancing requirements [36]. But not only parents' mental health was affected by lockdown measures.

Another consequence of lockdown measures is an increase in internalizing and externalizing problems for adolescents [3], school children [6] and even preschool children [4, 5]. For preschool children, the increase in parental stress and parental depressive and anxious symptomatology explained a large portion of the increases in children's own internalizing and externalizing problems during the COVID-19 pandemic [39–42]. Also longitudinal work by Rakickienė et al. [40] demonstrated that among screen time, physical activity and parental stress, parental stress was the only significant predictor of preschool children's internalizing and externalizing problems during lockdown. Concerning parental mental health, Dollberg et al. [41] were able to show that mothers' anxiety symptoms mediated the effect of the pandemic on preschool children's internalizing and externalizing problems. Also Frigerio et al. [42] found that maternal internalizing symptoms (mood symptoms) were associated with preschool children's internalizing (emotional reactive, anxious-depressed, withdrawn) problems and externalizing (aggressive) problems during the pandemic. Finally, Wang et al. [43] identified that higher parental well-being was related to lower mental health problems in preschool children during the pandemic. Overall, parental stress and parental mental health appear to explain important variance in the increase of internalizing and externalizing problems in preschool children during the pandemic.

Despite several studies examining the mental health of preschool children in light of the pandemic, a majority have been conducted during the first lockdown measures and at the beginning of the COVID-19 pandemic in 2020. However, the pandemic has been ongoing and several measures have been adjusted for follow-up lockdowns. In Germany, two nationwide lockdowns have taken place: a first one between March 2020 and May 2020 and a second one between December 2020 and May 2021. Both lockdowns were characterized by contact restrictions and shop, restaurant, school, and preschool closures. Regarding preschool closures, there was no replacement care available for preschool children except for a system called 'emergency care'. 'Emergency care' was offered at the preschools themselves and only available if parents worked in the healthcare system. An important difference between the two lockdowns was that during the second nationwide lockdown, criteria for using 'emergency care' were extended and parents that had no other options of caring for their child were allowed to take advantage of it. Broadening the access to preschool may have been an important choice for the development of young children. Preschool is a crucial time in the development of young children which enables them to interact with peers and build social skills [44, 45]. Furthermore, preschool attendance is a strong predictor for later academic, social and economic success [46]. Also for parents, preschool can be an aid in reducing parental stress by having more time to navigate daily struggles, to improve finances through longer work hours or to take more time for themselves [47].

Prior work focusing on the impact of COVID-19 measures on preschool children's mental health has been mostly concerned with risk factors and did not explicitly examine possible protective factors. Attending preschool may constitute a protective factor for the risk of increased internalizing and externalizing problems in children and deserves examination.

The purpose of the current study was to examine the effects of COVID-19 lockdown measures on internalizing and externalizing problems of preschool children, while considering the negative impact of parental stress, parental mental health, child attachment and particularly the positive impact of preschool attendance throughout the lockdown. Parents were questioned once about three different time points: two weeks before the second nationwide lockdown (T1), two weeks during the most critical phase of the second nationwide lockdown (T2) and the most recent two weeks (after the second lockdown) (T3). T1 and T2 were assessed retrospectively. It was hypothesized that increased parental stress, higher parental depressive and anxious symptomatology and child attachment problems at T1 and T2 would significantly predict children's internalizing and externalizing symptomatology at T2 and T3, respectively. Furthermore, it was expected that duration of preschool attendance per week throughout the lockdown would be a negative predictor of children's internalizing and externalizing symptomatology at T2 and T3.

## 2. Methods and materials

The current study was conducted as an online survey via the software tool PsyToolKit [48, 49]. Inclusion criteria were a) being the parent of a child between 2 and 6 years of age and b) preschool attendance of the child prior to the onset of the COVID-19 pandemic. Before filling out the survey, parents confirmed via checkboxes that they have read and understood their rights in relation to the study. Digital informed consent was obtained by an additional checkbox that had to be explicitly marked. The survey had a duration of approximately 45 minutes and after completion parents had the possibility to enter a valid email address and receive monetary compensation for participation in addition to a raffle of 20 vouchers of 25 euros each among participants. Email addresses were voluntarily entered into a separate survey and could not be matched to answers within the main survey. Data was collected between June 2021 and February 2022 after the second nationwide lockdown in Germany. The study was approved by the Ethics Committee of the University of Regensburg (20-1916-101) and preregistered in the German Clinical Trials Register (DRKS; DRKS00023812).

### 2.1. Participants

Participating parents were recruited via local preschools, flyers at supermarkets, notices on social media platforms and parent groups. Regarding local preschools, educators received flyers that were printed out and distributed among parents or sent digitally over parent mailing lists including a link to the survey and a QR code. A total of 53 preschools for children between 0–3 years and 67 preschools for children between 3–6 years were contacted. Out of these preschools 69 (57.50%) agreed to participate by handing out flyers or sending recruitment emails. Overall, $N = 128$ parents (112 mothers; 87.50 %) were included in the final sample. An additional $n = 16$ parents provided informed consent to participate in the survey, but did not enter any data. Out of the 128 data sets, 113 data sets were complete including all items and questionnaires (88.30 %).

### 2.2. Measures

Constructs measured were children's internalizing and externalizing problems, children's attachment problems, parental stress, parental depressive symptomatology, and parental anxious symptomatology. Additionally, demographic variables such as children's age and sex, parental age and sex, parental education, employment, number of people in the household, relationship status, financial situation, and hours per week that children attended emergency preschool during the lockdown were assessed.

**2.2.1. Internalizing and externalizing problems.** Children's internalizing and externalizing problems were measured via the *Strengths and Difficulties Questionnaire* (SDQ) [50]. The SDQ is a validated 25-item parent-report questionnaire with 5 scales: emotional problems, conduct problems, hyperactivity / inattention, peer relationship problems and prosocial behavior. The first four scales can be added for a total score. The prosocial behavior scale is a separate measure. Additionally, the scales emotional problems and peer relationship problems can be added into an internalizing score and the scales conduct problems and hyperactivity / inattention can be added into an externalizing score. The computation of an internalizing and an externalizing score has been shown to be the more reliable than the separate scales in community samples [51]. Reported internal consistency for the German parent-report version of the SDQ is good (Cronbach's α = 0.82 [52]). Scores above 7 on the internalizing and externalizing scale are considered at-risk and scores above 9 are considered critical. The SDQ was the main measure for internalizing and externalizing problems. Additional questionnaires were included in order to validate newer questionnaires and to confirm construct validity.

Additionally, internalizing problems were examined via two more parent-report questionnaires, the *Child Behavior Checklist 1.5–5* (CBCL) [53] and the *Children's Moods, Fears and Worries Questionnaire* (CMFWQ) [54]. The CBCL is a validated 100-item parent-report questionnaire with the following scales: aggressive behavior, anxious / depressed, attention problems, rule-breaking behavior, somatic complaints, social problems, thought problems and withdrawn / depressed. A score for internalizing problems can be computed by adding the scales anxious / depressed, withdrawn / depressed, and somatic complaints scales. In the present study, only items relevant for computing the internalizing score were included. The German version of the CBCL has demonstrated good internal consistency (Cronbach's α > 0.89 [55]).

The CMFWQ is a relatively new parent-report questionnaire with three versions depending on the age of the child. A version for 2-year-olds, 4-year-olds and 7-year-olds exists with 35, 38 and 34 items each. As anxiety and depression are difficult to discern in preschool children [56], the CMFWQ was developed to enable a sensitive measure for early signs of internalizing problems in community samples [54]. It is more liberal in symptoms and less based on diagnostic criteria of anxiety and depression than the SDQ and the CBCL. In English, the CMFWQ demonstrated good internal consistency in community and at-risk samples (Cronbach's α > 0.91 [16, 57]). Also a clinical cut-off was determined with scores above 2.87 showing the best sensitivity and specificity for internalizing disorders [58]. For the current study, the CMFWQ was translated into German according to best-practice recommendations with two back-translations and in interaction with the original author. Internal consistency in the present sample was very good with Cronbach's α = 0.94. The CMFWQ and the CBCL were administered additionally in order to validate the CMFWQ for future use.

All three measures were administered in relation to three time points, 1) two weeks before the nationwide lockdown (T1), 2) two weeks during the most difficult phase of the lockdown (T2), and 3) the most recent two weeks after the lockdown (T3). See Fig 1 for on overview of questionnaires administered across T1, T2 and T3.

### 2.2.2. Attachment problems

Attachment problems were measured via the short form of the *Relationship Problems Questionnaire* (RPQ) [59]. The RPQ is a 10-item parent-report questionnaire developed as a screening tool for attachment disorder. It works well in clinical [60] as well as community samples and has demonstrated good internal consistency (Cronbach's α = 0.85 [61]). Also the German version has been validated, showed good internal consistency (Cronbach's α = 0.82) and

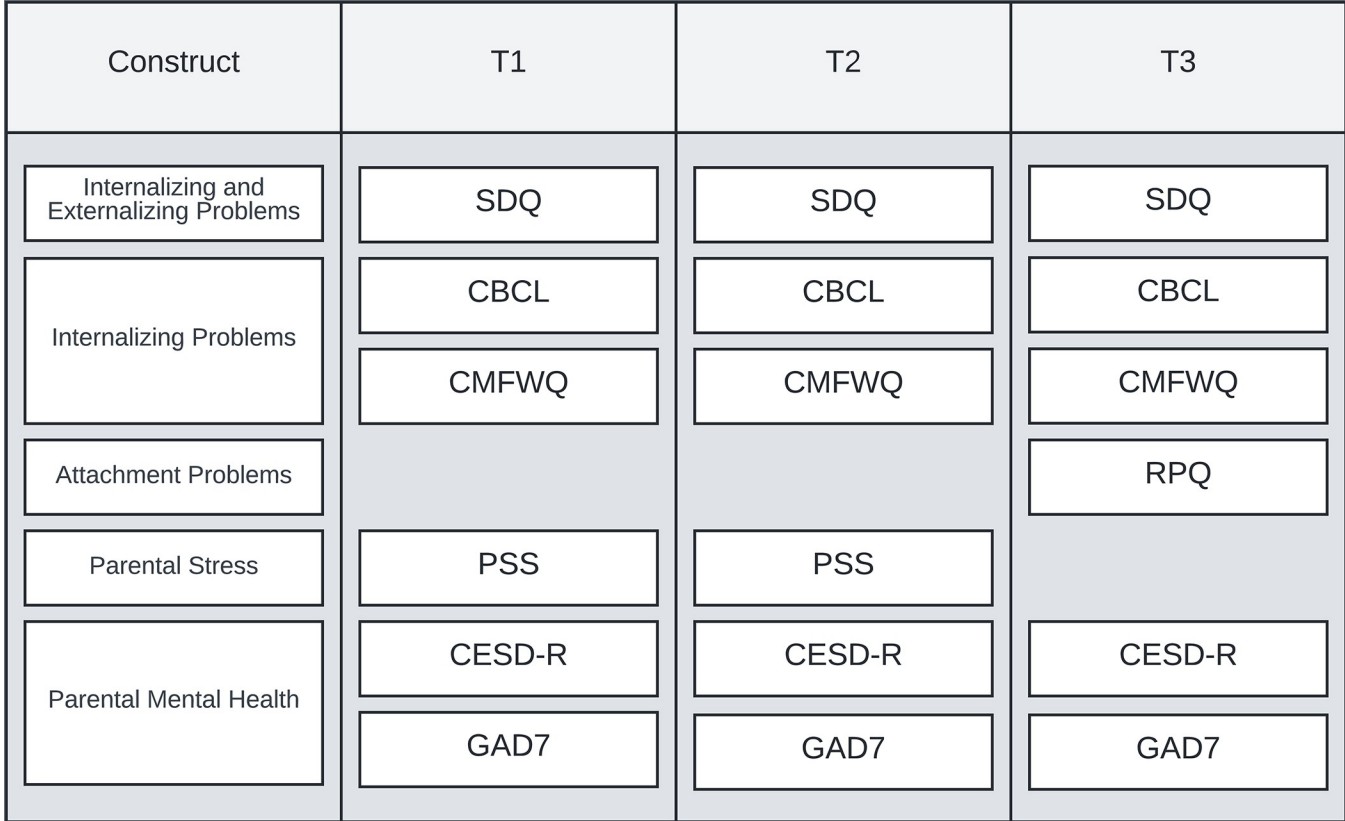

**Fig 1. Overview of measures across the three time points.** *Note.* SDQ = Strengths and Difficulties Questionnaire, CMFWQ = Children's Moods Fears and Worries Questionnaire, CBCL = Child Behavior Checklist, CMFWQ = Children's Moods, Fears and Worries Questionnaire, RPQ = Relationship Problems Questionnaire, CESD-R = Center for Epidemiologic Studies Depression Scale Revised, GAD7 = Generalized Anxiety Disorder 7, PSS = Parental Stress Scale (PSS). T1 = two weeks before the lockdown, T2 = two weeks during the most difficult time during the lockdown, T3 = the most recent two weeks after the lockdown.

determined a cut-off score of 4.5 [62]. In the present study, the RPQ was used dimensionally as an approximation for problems in children's attachment style. The RPQ was only administered in relation to T3.

**2.2.3. Parental stress.** Parental stress was assessed via the *Parental Stress Scale* (PSS) [63]. The PSS is an 18-item questionnaire inquiring about parents' stress in relation to their role as a parent. The questionnaire considers both positive (i.e., I feel close to my children) and negative aspects of parenthood (i.e., caring for my child(ren) sometimes takes more time and energy than I have to give). The English original demonstrated good internal consistency (Cronbach's α = 0.83 [63]), also across various samples [64–66]. The German version translated by Kölch & Schmid [67] has also demonstrated good internal consistency (Cronbach's α = 0.83). Parental stress was assessed for the time points T1 and T2.

**2.2.4. Parental mental health.** Parental mental health was assessed through a measure for depressive symptomatology, the *Center for Epidemiologic Studies Depression Scale Revised* (CESD-R [68], revised by Eaton et al. [69]) and a measure for anxious symptomatology, the *Generalized Anxiety Disorder 7* (GAD-7) [70]. The CESD-R is a 20-item self-report questionnaire that assesses depression on nine different scales: Sadness (Dysphoria), Loss of Interest (Anhedonia), Appetite, Sleep, Thinking / Concentration, Guilt (Worthlessness), Tired (Fatigue), Movement (Agitation) and Suicidal Ideation. A total score can be computed across all items which ranges between 0 and 60 points. A score of less than 16 points indicates no

clinical significance whereas a score above 16 suggests a subthreshold of depression. Based on the scales, DSM-5 diagnoses can be determined, however, in the present study the CESD-R was used as a dimensional measure of symptomatology. The original CESD-R has demonstrated good internal consistency (Cronbach's α = 0.82 [71]). Internal consistency of the CESD-R was good in the present sample with Cronbach's α = 0.93 for T3.

The GAD-7 is a brief 7-item self-report questionnaire, developed to assess generalized anxiety disorder in patients. However, the questionnaire has also been shown to be reliable in identifying general anxious symptomatology in clinical samples [72], as well as community samples [73]. Scores can range between 0 and 21 with 0–4 indicating low, 5–9 indicating mild, 10–14 indicating medium and 15–21 indicating high anxious symptomatology. Internal consistency has been found to be good across all samples (Cronbach's α = 0.89 [73]).

## 2.3. Statistical analysis

Statistical analyses were performed using the R statistical package, version 4.0.2 [74] for linear mixed effect models (LMEs) and within-group comparisons and SPSS 28 [75] for all other analyses. An a priori power analysis was computed via G*Power [76] to determine the appropriate sample size for a linear regression predicting children's internalizing and externalizing problems. For a linear regression with a medium effect size of $f^2$ = .15 and six predictors, a total sample size of $n$ = 98 would be necessary in order to achieve a power of .80. Although prior work found higher effect sizes [40], a medium effect size was chosen to keep the potential for higher power within the calculation. Six predictors were chosen: parental depressive and anxious symptomatology, children's attachment problems, parental stress, hours of preschool attendance per week and possible additional demographic variables.

In a first step, the development of children's internalizing and externalizing problems, parental depressive and anxious symptomatology and parental stress were examined over the queried time points. As the main variables were not normally distributed, Friedman's test was used to determine whether a significant change between T1, T2, and T3 took place, followed up by post-hoc tests for the specific relationships. In a second step, the impact of demographic variables such as age, sex, parental relationship status, presence of siblings, and parental education was examined via Mann-Whitney $U$ tests and bivariate correlations. Also, correlations between the main variables were computed to determine meaningful relationships.

In order to predict preschooler's internalizing and externalizing problems at T2, linear regressions were computed with parental stress at T1, parental anxious and depressive symptomatology at T1, hours of preschool attendance per week throughout the lockdown, and additional demographic variables that were identified previously as independent variables. Additionally, internalizing, and externalizing problems at T3 were predicted using parental stress at T2, parental anxious and depressive symptomatology at T2, children's attachment problems, hours of preschool attendance per week, and once again additional demographic variables. Models were computed separately for internalizing and externalizing problems.

Finally, in order to determine variable relationships across the time points, LMEs were computed with internalizing and externalizing problems as the dependent variable and parental stress, parental depressive and anxious symptomatology, children's attachment problems, hours of preschool attendance per week and additional demographic aspects as independent variables. Subject was added as a random effect to consider differences over the three assessed periods. LMEs have the advantage of robustness against missing values, unequal sample sizes and violations of normality within the dependent variable [77]. Models were computed using the *lme4* package in R [78] including the *lmerTest* package for *p*-values according to the Satterthwaite approximation to degrees of freedom [79]. The package *r2glmm* was used to

determine $R^2$ for fixed effects [80], including the semi-partial (marginal) $R^2$ along with confidence intervals. According to best practice recommendations [81] a null model with a random intercept was computed and compared to a model including all predictors (maximized model). To achieve a parsimonious model, a reduced model including only significant predictors was computed and compared to the maximized model. The model with the lowest model indices (Akaike Information Criterion (AIC [82]); Bayesian Information Criterion (BIC [83]); log likelihood ratio) that was significantly different from the null model was chosen.

The false-discovery rate (FDR [84]) was applied to correct for multiple comparisons where appropriate and the *p*-value was set to .05.

## 3. Results

### 3.1. Sample characteristics

A total of $N$ = 128 parents participated in the online survey (112 mothers, 16 fathers). Mothers were 36.31 years old ($SD$ = 4.43) and fathers were 40.27 years old ($SD$ = 5.73) on average. A total of 50.80 % of parents had a university degree, 26.60 % had a high school degree, 12.50 % finished vocational school, 6.30% finished 10 years of high school education and 3.90 % finished 9 years of high school education. Among parents, 4.70 % were single, 91.40 % were married or lived with their partner and 3.90 % were divorced or widowed. Parents were employees (82.80 %), freelancers (2.30 %), civil servants (10.90 %) and students (3.90 %). At the time of the survey, 20.30 % were working fulltime, 60.90 % were working part time and 18.80% were not working. A total of 86.70 % of families were monolingual.

Children were 4.17 years old on average ($SD$ = 1.12) with 8 2-year-olds (6.30 %), 31 3-year-olds (24.20 %), 35 4-year-olds (27.30 %), 39 5-year-olds (30.50 %) and 15 6-year-olds (11.70 %). Among the children 54 (42.20 %) were female, 49 (38.30 %) were male and 25 were missing information on sex due to an error in the first version of the survey (19.50%). About 22.70 % of children had no siblings, 39.10 % had older siblings, 32.80 % had younger siblings and 5.50 % had both older and younger siblings. During the nationwide lockdown, 57.00 % of children continuously attended emergency preschool services, whereas 43.00 % of children were supervised at home. Preschool attendance ranged from 8 hours per week to 50 hours per week with a mean attendance of 26.22 hours ($SD$ = 10.77 hours).

Percentage of children and parents scoring above clinical cut-offs across the three time points are depicted in Table 1.

### 3.2. Change over time

Changes in the main variables of interest over the three assessed time points were examined via Friedman's tests, followed by FDR-corrected post-hoc tests. See Fig 2 for a graphical depiction of changes over time for children's internalizing and externalizing problems (SDQ (Fig 2A and 2B); CMFWQ (Fig 2C)), parental depressive and anxious symptomatology (CESD-R (Fig 2D); GAD7 (Fig 2E)), and parental stress (PSS; Fig 2F), including significant post-hoc tests. All variables of interest showed significant changes over time, suggesting increases from the time before the lockdown (T1) towards the most difficult time of the lockdown (T2). Furthermore, decreases from T2 towards the time of the survey (after lockdown; T3) could be observed, while still showing significantly higher values at T3 compared to T1. An exception are children's externalizing problems that remained high between T2 and T3, suggesting no reduction after the lockdown. Parental stress was only examined at T1 and T2, showing a significant increase from an average of 36.93 at T1 to an average of 43.43 at T2.

**Table 1. Overview of children and parents scoring above clinical cut-offs.**

| | Measure | T1 | T2 | T3 |
|---|---|---|---|---|
| **Children** | **SDQ** | N (%) | N (%) | N (%) |
| | Internalizing | 125 | 125 | 125 |
| | *at risk* | 4 (3.20 %) | 7 (5.60 %) | 7 (5.60 %) |
| | *critical* | 5 (3.90 %) | 19 (15.20 %) | 15 (12.00 %) |
| | Externalizing | 125 | 125 | 125 |
| | *at risk* | 31 (24.80 %) | 23 (18.40 %) | 21 (16.80 %) |
| | *critical* | 32 (25.60 %) | 49 (39.20 %) | 46 (36.80 %) |
| | Total score | 125 | 125 | 125 |
| | *at risk* | 19 (15.20 %) | 16 (12.80 %) | 15 (12.00 %) |
| | *critical* | 8 (6.40 %) | 27 (21.60 %) | 21 (16.80 %) |
| | **CMFWQ** | | | |
| | Total score | 128 | 128 | 128 |
| | > 2.87 | 3 (2.30 %) | 18 (14.10 %) | 15 (11.70 %) |
| | **RPQ** | | | |
| | Total score | - | - | 124 |
| | > 5 | | | 30 (24.20 %) |
| **Parents** | **CESD-R** | | | |
| | Total score | 113 | 113 | 113 |
| | > 16 | 18 (15.90 %) | 54 (47.80 %) | 34 (30.10 %) |
| | **GAD7** | 113 | 113 | 113 |
| | mild | 26 (23.00 %) | 39 (34.50 %) | 35 (31.00 %) |
| | moderate | 11 (9.70 %) | 19 (16.80 %) | 13 (11.50 %) |
| | severe | 2 (1.80 %) | 13 (11.50 %) | 7 (6.20 %) |

*Note.* SDQ = Strengths and Difficulties Questionnaire, CMFWQ = Children's Moods, Fears and Worries Questionnaire, RPQ = Relationship Problems Questionnaire, CESD-R = Center for Epidemiologic Studies Depression Scale Revised, GAD7 = Generalized Anxiety Disorder 7. The RPQ was only assessed for T3. T1 = two weeks before the lockdown, T2 = two weeks during the most difficult time during the lockdown, T3 = the most recent two weeks after the lockdown. Parental stress is not reported as no cut-offs were available.

### 3.3. Demographic variables

The impact of demographic variables on children's internalizing and externalizing problems was examined for children's age, parental age, sex, parental relationship status, presence of siblings, and parental education. The SDQ was used as the main measure for internalizing problems. There was no correlation of children's or parental age with either internalizing or externalizing problems ($\tau < .10$, $p > .05$). However, parental education was significantly correlated with children's externalizing problems at T1, T2 and T3 (T1: $\tau = -.26$, $p < .001$; T2: $\tau = -.31$, $p < .001$; T3: $\tau = -.30$, $p < .001$). There were also no sex differences for internalizing and externalizing problems across the three time points ($U = 1017.00$–$1285.00$, $p > .05$). Also, parental relationship and presence of siblings had no significant impact on children's internalizing and externalizing problems in a Kruskal-Wallis test ($H = .09$–$4.81$, $p > .05$).

### 3.4 Correlations

Correlations among the main variables of interest (children's internalizing, externalizing and attachment problems; parental stress; parental anxious and depressive symptomatology and hours of emergency preschool attendance per week) are depicted in Table 2. All measures of internalizing problems in children (SDQ, CBCL, CMFWQ) correlated significantly with each other ($\tau = .35$

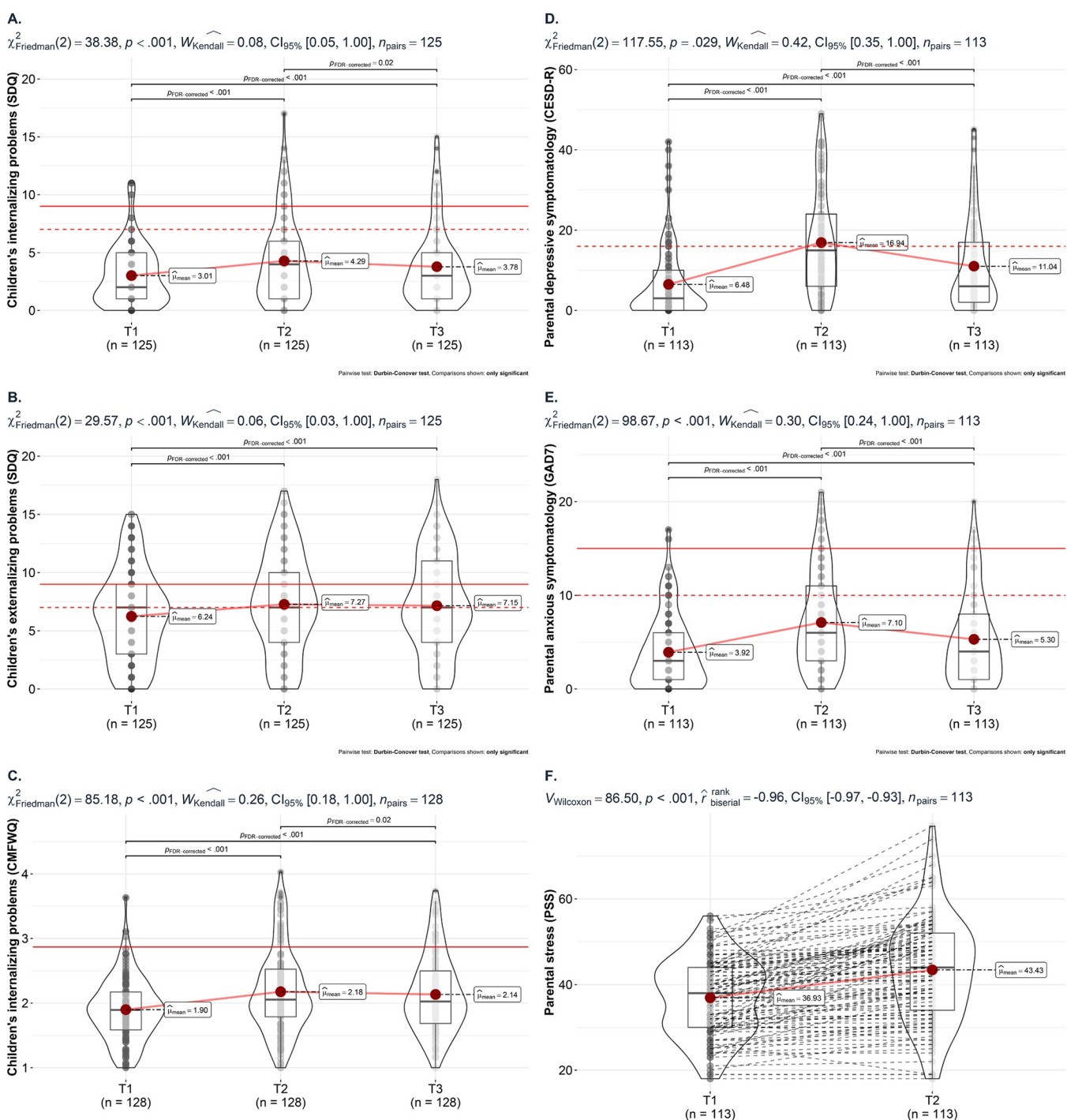

**Fig 2. Changes in children's and parents' psychopathology over time.** *Note.* SDQ = Strengths and Difficulties Questionnaire, CMFWQ = Children's Moods Fears and Worries Questionnaire, CESD-R = Center for Epidemiologic Studies Depression Scale Revised, GAD7 = Generalized Anxiety Disorder 7, PSS = Parental Stress Scale (PSS). Dashed red lines refer to at risk scores and filled red lines refer to clinical cut off scores. T1 = two weeks before the lockdown, T2 = two weeks during the most difficult time during the lockdown, T3 = the most recent two weeks after the lockdown. Parental stress is not reported as no cut-offs were available.

**Table 2. Correlations among the main variables of interest (children's internalizing, externalizing and attachment problems; parental stress, parental depressive and anxious symptomatology and hours of emergency preschool attendance).** Shade of gray represents strength of correlation with darker tones indicating stronger relationships.

| | 1. | 2. | 3. | 4. | 5. | 6. | 7. | 8. | 9. | 10. | 11. | 12. | 13. | 14. | 15. | 16. |
|---|---|---|---|---|---|---|---|---|---|---|---|---|---|---|---|---|
| 1. Preschool attendance | | .02 | -.15 | -.21* | -.20* | -.13 | -.18* | -.13 | -.07 | -.19** | -.14 | .03 | -.09 | -.01 | -.02 | -.07 |
| 2. RPQ | | | .18* | .26*** | .24*** | .34*** | .36*** | .35*** | .25*** | .23*** | .32*** | .22* | .27** | .29** | .22* | .28** |
| 3. SDQ Inter T1 | | | | .66*** | .68*** | .19* | .19* | .21*** | .14 | .21* | .24** | .12 | .17 | .16 | .14 | .25** |
| 4. SDQ Inter T2 | | | | | .80*** | .21* | .31*** | .29*** | .18** | .35*** | .39*** | .17 | .21** | .32*** | .21* | .37*** |
| 5. SDQ Inter T3 | | | | | | .23*** | .29*** | .32*** | .15 | .26** | .35*** | .12 | .24** | .26** | .13 | .24** |
| 6. SDQ Exter T1 | | | | | | | .75*** | .73*** | .23* | .15* | .26** | .20* | .14 | .25** | .28** | .23** |
| 7. SDQ Exter T2 | | | | | | | | .85*** | .18* | .29** | .35*** | .18* | .26** | .32*** | .23** | .30*** |
| 8. SDQ Exter T3 | | | | | | | | | .16 | .24** | .34*** | .19* | .25** | .31*** | .23** | .28** |
| 9. CESD-R T1 | | | | | | | | | | .43*** | .48*** | .53*** | .34*** | .35*** | .38*** | .30*** |
| 10. CESD-R T2 | | | | | | | | | | | .55*** | .41*** | .62*** | .42*** | .22** | .44*** |
| 11. CESD-R T3 | | | | | | | | | | | | .40*** | .46*** | .61*** | .28** | .41*** |
| 12. GAD7 T1 | | | | | | | | | | | | | .61*** | .64*** | .35*** | .39*** |
| 13. GAD7 T2 | | | | | | | | | | | | | | .63*** | .24** | .45*** |
| 14. GAD7 T3 | | | | | | | | | | | | | | | .31*** | .41*** |
| 15. PSS T1 | | | | | | | | | | | | | | | | .61*** |
| 16. PSS T2 | | | | | | | | | | | | | | | | |

Note. RPQ = Relationship Problems Questionnaire, SDQ = Strengths and Difficulties Questionnaire, CESD-R = Center for Epidemiologic Studies Depression Scale Revised, GAD7 = Generalized Anxiety Disorder 7.

\* $p < .05$

\*\* $p < .01$

\*\*\* $p < .001$. $p$-values are FDR-corrected. T1 = two weeks before the lockdown, T2 = two weeks during the most difficult time during the lockdown, T3 = the most recent two weeks after the lockdown.

- .59, $p < .001$). In the main analyses, only the SDQ internalizing and externalizing scales were used as outcome variables as the CMFWQ and the CBCL were included for validation.

### 3.5. Predicting internalizing and externalizing problems at T2

In order to predict children's internalizing and externalizing problems at T2, two linear regressions were computed with the independent variables parental stress at T1, parental anxious and depressive symptomatology at T1, and hours of preschool attendance per week throughout the lockdown. For externalizing problems, parental education was added as an additional demographic variable. The regression model predicting internalizing problems was significant ($F(4,108) = 5.86$, $p < .001$) and explained 17.80 % of the variance in children's internalizing problems at T2. Only preschool attendance was a significant predictor with an increase of 1 hour preschool attendance resulting in a reduction of 0.08 points on the SDQ (see Table 3).

The regression model predicting externalizing problems was also significant ($F(5,107) = 10.73$, $p < .001$) and explained 33.40 % of the variance in children's externalizing problems at T2. Significant predictors were parental stress at T1, preschool attendance and parental education, with one-point increases resulting in a 0.18-point increase and in 0.05- and 1.51-point decreases in externalizing problems at T2, respectively (see Table 3).

### 3.6. Predicting internalizing and externalizing problems at T3

Linear regression models were also computed for internalizing and externalizing problems at T3. Once again, parental anxious and depressive symptomatology at T2, parental stress at T2

**Table 3. Results of linear regression models predicting children's internalizing and externalizing problems at T2 and T3.**

| Dependent Variable | Predictor | B | SE | β | t | p | $R^2$ |
|---|---|---|---|---|---|---|---|
| Internalizing problems at T2 | Parental anxious symptomatology at T1 | .23 | .14 | .23 | 1.66 | .099 | .18 |
| | Parental depressive symptomatology at T1 | -.02 | .06 | -.04 | -.28 | .780 | |
| | Parental stress at T1 | .06 | .04 | .14 | 1.36 | .176 | |
| | Preschool attendance throughout lockdown | -.08 | .02 | -.31 | -3.48 | < .001 | |
| Externalizing problems at T2 | Parental anxious symptomatology at T1 | .09 | .14 | .08 | .66 | .512 | .33 |
| | Parental depressive symptomatology at T1 | .00 | .06 | .01 | .05 | .958 | |
| | Parental stress at T1 | .18 | .05 | .36 | 3.92 | < .001 | |
| | Preschool attendance throughout lockdown | -.05 | .02 | -.17 | -2.10 | .038 | |
| | Parental education | -1.51 | .31 | -.40 | -4.86 | < .001 | |
| Internalizing problems at T3 | Parental anxious symptomatology at T1 | .14 | .10 | .20 | 1.35 | .181 | .19 |
| | Parental depressive symptomatology at T1 | .02 | .04 | .08 | .52 | .604 | |
| | Parental stress at T1 | .03 | .03 | .09 | .77 | .441 | |
| | Preschool attendance throughout lockdown | -.05 | .02 | -.22 | -2.44 | .016 | |
| Externalizing problems at T3 | Parental anxious symptomatology at T1 | .12 | .11 | .14 | 1.02 | .308 | .34 |
| | Parental depressive symptomatology at T1 | -.04 | .05 | -.13 | -.91 | .364 | |
| | Parental stress at T1 | .16 | .04 | .45 | 4.31 | < .001 | |
| | Preschool attendance throughout lockdown | -.02 | .02 | -.08 | -1.01 | .317 | |
| | Parental education | -1.41 | .31 | -.37 | -4.51 | < .001 | |

*Note.* SDQ (= Strengths and Difficulties Questionnaire) scales were used as outcome variables. T1 = two weeks before the lockdown, T2 = two weeks during the most difficult time during the lockdown, T3 = the most recent two weeks after the lockdown.

and preschool attendance were added as independent variables. Parental education was added as an independent variable to the model predicting externalizing problems at T3.

The regression for internalizing problems at T3 was significant ($F(4,108) = 6.45, p < .001$) and explained 19.30 % of the variance in children's internalizing problems. Once again, only preschool attendance was a significant predictor with a 1-hour increase of preschool attendance resulting in a 0.05-point reduction in internalizing problems (see Table 3).

The regression model predicting externalizing problems at T3 was also significant ($F(5,107) = 11.18, p < .001$) and explained 34.30 % of variance in children's externalizing problems. Significant predictors were parental stress and parental education, with 1-point increases resulting in 0.02-point and 1.41-point decreases in externalizing problems (see Table 3).

### 3.7. Predicting internalizing and externalizing problems over time

In order to examine predictors of children's internalizing and externalizing problems across the three time points, linear mixed effect models were computed. Internalizing and externalizing problems were set as the dependent variables and time, parental stress, parental depressive and anxious symptomatology, children's attachment problems, hours of preschool attendance per week and parental education as independent variables. Subject was added as a random effect to ensure individual variance over the three time points was considered. Children's attachment problems were only assessed at T3 and could not be added as a predictor to the regression models, however, in the linear mixed effect model it could be added as a predictor. Three models were computed, one for total problems (internalizing + externalizing) and one for internalizing and externalizing separately. Null models, maximized models and reduced models for the three dependent variables are depicted in Table 4. For total problems and externalizing problems, the reduced model showed the best fit indices while being significantly

**Table 4. Model comparisons of linear-mixed effect models predicting internalizing and externalizing problems.**

| Outcome Variable | Model | AIC | BIC | logLik | $R^2$ | Vs. Null Model | | Vs. Max Model | |
|---|---|---|---|---|---|---|---|---|---|
| | | | | | | $\chi^2$ | p-value | $\chi^2$ | p-value |
| Total problems (SDQ) | Null Model | 1392.90 | 1403.20 | -693.47 | .00 | | | | |
| | Max Model | 1219.30 | 1253.50 | -599.66 | .55 | 187.62 | < .001 | | |
| | Reduced Model | 1216.30 | 1243.70 | -600.17 | .56 | 186.50 | < .001 | 1.02 | .600 |
| Internalizing problems (SDQ) | Null Model | 1129.50 | 1139.70 | -561.73 | .00 | | | | |
| | Max Model | 1027.80 | 1058.60 | -504.90 | .36 | 113.65 | < .001 | | |
| | Reduced Model | 1028.70 | 1049.30 | -508.37 | .34 | 106.72 | < .001 | 6.93 | .074 |
| Externalizing problems (SDQ) | Null Model | 1149.70 | 1160.00 | -571.84 | .00 | | | | |
| | Max Model | 1015.60 | 1049.80 | -497.78 | .44 | 148.12 | < .001 | | |
| | Reduced Model | 1012.40 | 1036.30 | -499.19 | .43 | 145.30 | < .001 | 2.81 | .421 |

*Note*. The null model refers to a random intercept only model. The max model refers to a model including all relevant predictors and the reduced model only includes significant predictors from the max model. SDQ = Strengths and Difficulties Questionnaire.

different from the null model. However, for internalizing problems the maximized model showed the better fit indices, even if the reduced and the maximized model did not differ significantly from each other.

Overall, the models explained 55.60 %, 34.40 % and 42.90 % of the variance for the total problems score, the internalizing score, and the externalizing score, respectively. The total problems score was significantly predicted by parental stress, parental anxious symptomatology, children's preschool attendance throughout the lockdown, children's attachment problems and parental education. For the internalizing problems score, parental stress, parental anxious symptomatology, children's preschool attendance throughout the lockdown and children's attachment problems were significant predictors over time. Finally, parental stress, parental anxious symptomatology, children's attachment problems and parental education were significant predictors of externalizing problems. See Table 5 for a depiction of the three models and fixed effects.

**Table 5. Main fixed effects of interest within LMEs predicting internalizing and externalizing problems.**

| | Fixed Effect | Estimate | SD | t | p | $R^2_{\beta*}$ | $R^2_{\beta*}$ CI |
|---|---|---|---|---|---|---|---|
| Total problems (SDQ) | Parental stress | .21 | .03 | 6.61 | < .001 | .15 | .08 - .24 |
| | Parental anxious symptomatology | .32 | .07 | 4.64 | < .001 | .07 | .02 - .15 |
| | Preschool attendance throughout lockdown | -.06 | .03 | -2.63 | < .001 | .05 | .01 - .12 |
| | Children's attachment problems | .47 | .12 | 3.99 | < .001 | .10 | .04 - .19 |
| | Parental education | -1.28 | .35 | -3.68 | < .001 | .09 | .04 - .17 |
| Internalizing problems (SDQ) | Time | -.17 | .12 | -1.44 | .153 | .00 | .00 - .04 |
| | Parental stress | .08 | .02 | 3.44 | < .001 | .06 | .01 - .11 |
| | Parental depressive symptomatology | .01 | .02 | .59 | .553 | .00 | .00 - .03 |
| | Parental anxious symptomatology | .15 | .06 | 2.39 | .018 | .02 | .00 - .07 |
| | Preschool attendance throughout lockdown | -.04 | .02 | -2.59 | .010 | .05 | .01 - .13 |
| | Children's attachment problems | .16 | .07 | 2.23 | .027 | .03 | .00 - .10 |
| Externalizing problems (SDQ) | Parental stress | .10 | .02 | 5.26 | < .001 | .07 | .02 - .15 |
| | Parental anxious symptomatology | .15 | .04 | 3.70 | < .001 | .03 | .00 - .09 |
| | Children's attachment problems | .35 | .09 | 4.00 | < .001 | .11 | .04 - .19 |
| | Parental education | -1.10 | .26 | -4.25 | < .001 | .13 | .06 - .21 |

*Note*. For the total problems score and the externalizing score the reduced model is reported. For internalizing problems, the maximized model is reported.

## 4. Discussion

The aim of this study was to examine the effects of COVID-19 lockdown measures on preschool children's internalizing and externalizing problems, while considering the effect of parental stress, parental mental health, child attachment, and the option of preschool attendance throughout the second nationwide lockdown in Germany. Results showed a rapid increase in children's internalizing and externalizing problems from the time before the lockdown (T1) to the time of the lockdown (T2), while also remaining high after the lockdown (T3). Also, parental anxious and depressive symptomatology, as well as parental stress increased rapidly over the time of the lockdown. Finally, possible predictors for internalizing and externalizing problems over the examined time could be identified. Internalizing problems were positively predicted by parental stress, parental anxious symptomatology, and children's attachment problems and negatively predicted by children's emergency preschool attendance during the lockdown. Externalizing problems were positively predicted by parental stress, parental anxious symptomatology, children's attachment problems and negatively predicted by parental education.

The impact of lockdown measures on children's internalizing and externalizing problems becomes apparent when considering the percentage of children scoring above clinical cut-offs. Whereas only 1 in 15 children received a critical score on the SDQ before the lockdown (T1), this number already changed to 1 in 5 during the lockdown (T2) and remained at 1 in 8 after the lockdown (T3). Increases in SDQ scores following the start of the pandemic have also been identified by Specht et al. [5], however, in the examined sample in Denmark only an increase of 5 % in scores above the clinical cut-off could be observed. Differences in results compared to the present study may be due to the lockdown time examined (early 2020 vs. early 2021) and the duration of the lockdown (3 weeks vs. 5 months). Similarly to general lockdown measures [85], preschool closures as well as social distancing measures are likely to have a greater impact on mental health, the longer they are in place. Also, Christner et al. [86] and Cantiani et al. [4] report increases in internalizing and externalizing problems throughout early lockdown measures in 2020, but the percentage of children scoring above clinical cut-offs is unfortunately not reported. Overall, it can be concluded that increases in internalizing and externalizing problems in preschool children can not only be observed in response to nationwide lockdowns early in the pandemic, but also for (less restrictive) subsequent lockdowns. Additionally, the severity of internalizing and externalizing problems observed in the present study, may match the duration of the lockdown (5 months), instead of its restrictions (less strict compared to the first nationwide lockdown). Although the number of children scoring above clinical cut-off significantly reduces after the most difficult phase of the lockdown, it is not negligible and deserves further examination and, possibly, intervention.

Besides changes in internalizing and externalizing problems over the time of the second nationwide lockdown, possible predictors of internalizing and externalizing problems were examined. Similarly to previous studies during the early phase of the pandemic [39, 40, 86], parental stress explained a significant portion of variance for both internalizing and externalizing problems over time. Also parental mental health in the form of depressive and anxious symptomatology has been previously identified as a predictor of children's internalizing and externalizing problems [41, 42]. In the present study, parental anxious symptomatology, but not depressive symptomatology explained a significant portion of variance in both children's internalizing and externalizing problems. The effect was small to medium with an increase of 1 on the GAD-7 score resulting in an increase of 0.15 on the SDQ internalizing and externalizing scales. Frigerio et al. [42] identified an effect of parental depressive symptomatology on children's internalizing and externalizing problems during the COVID-19 pandemic.

However, children were younger on average (1 year old) than in the present study (4 years old), thus, the impact of maternal mood symptoms may have been stronger [30, 31]. Overall, the negative impact of parental stress and parental anxious symptomatology on preschool children's internalizing and externalizing problems could be confirmed for the second nationwide COVID-19 lockdown in Germany.

A protective parental factor that could be identified in the present study is parental education. Parental education explained a significant portion of variance in children's externalizing problems over the queried time. Previous studies have identified a negative relationship between socio-economic status and externalizing problems in preschool children [87, 88]. Specifically, parental education has been consistently identified as a negative predictor of internalizing and externalizing problems in preschool children in sufficiently powered samples [89]. A study by Zhang [90] has found that parental education was negatively associated with children's externalizing problems, even when controlling for mother-child or father-child conflict [90]. One of the reasons for the influence of parental education is argued to be that lower parental education is commonly associated with a lower socioeconomic status and, thus, with additional stress that requires resources to overcome [89]. What may be noted is the high level of education in the present sample. As preschool attendance can be costly in Germany, especially for children below the age of 3, high level of education is common among parents and remains representative.

In addition to parental factors, also children's attachment problems have been identified as significant predictors of internalizing and externalizing problems over time during the second nationwide lockdown. The impact of insecure attachment on internalizing problems in early childhood has been investigated and identified previously [32, 33], however, the role of attachment in relation to lockdown measures due to the COVID-19 pandemic has not been examined as of now. The present analysis found that children who scored higher on an attachment problems screening measure, were more likely to have higher internalizing and externalizing problem scores. Overall, attachment style has been shown to modulate the impact of the COVID-19 pandemic on the mental health of adults [91]; thus, it seems reasonable that the same relationship holds for children who have less mechanisms to compensate for these effects.

Finally, the present study examined the protective impact of preschool children's attendance of emergency preschool services during the second nationwide lockdown and assessed its role as a predictor for children's internalizing and externalizing problems over time. The second nationwide lockdown in Germany is an enabling period to examine this question, as more children were able to attend emergency preschool services and thus, differences between children attending and those not attending could be investigated. Analyses revealed preschool attendance during the lockdown as a negative predictor (protective factor) of internalizing problems, but not of externalizing problems. Here it is important to note that variability in preschool attendance was high with some children attending up to 50 hours per week. Several studies have shown the benefit of preschool attendance before the onset of the COVID-19 pandemic [44–46], among them improving social skills and later academic success. In the context of the COVID-19 pandemic, Cantiani et al. [4] have been able to show that contact with peers (through preschool) was a protective factor against internalizing and externalizing problems. However, keeping up contact with their peers is probably not the only reason for preschool having a protective effect on children. Additionally, preschool can serve as a relief to parents who can take time for themselves or increase work hours to improve their financial situation [47]. During nationwide lockdowns, many parents have switched to working from home and supervising preschool children during this period may have been an additional burden [92]. Preschool attendance during the nationwide lockdown may, thus, not only influence children's

internalizing and externalizing problems directly, but also serve to reduce parental stress. Parental stress, in turn, negatively impacts children's internalizing and externalizing problems. Overall, the findings emphasize the importance of children's preschool attendance during the pandemic and provide evidence against future restrictions involving preschool closures.

The present study has several strengths, among them a sufficiently powered sample of parents of preschool children, validated measures for a variety of constructs and a direct comparison of preschool children attending and not attending preschool during the lockdown. However, the study also had some limitations. First, two time points of the study (T1, T2) were assessed retrospectively. Retrospective questions about subjective feelings and mental health are known to be prone to recall bias, in particular participants tend to report the past more similar to the present [93]. Parents may thus have been biased in their answers based on their children's internalizing and externalizing problems at T3 and possibly underreported difficulties at T1 and T2. Although questions were kept short and clear and the time queried was not too long ago to improve recall [94], differences between pre- and post-pandemic values may have been underestimated. Future studies should use longitudinal designs to examine whether the observed increase in internalizing and externalizing problems remains constant or reduces along with COVID-19 restrictions. Parental stress has been examined previously and showed similar developments [95], but preschoolers problems have not been examined yet. An additional limitation was only using parent-report measures in order to assess internalizing and externalizing problems. Although the measures used show high validity and reliability [51, 96], parents may have been influenced by their own psychological burden and therefore overestimated children's problems. As very few reliable measures of internalizing and externalizing problems in preschool children are available, future studies may ask both parents as well as educators to fill out questionnaires in order to have a more reliable assessment. Furthermore, several measures used have not been validated for retrospective assessments. Although questionnaires have been developed to be precise and clear and showed good internal consistency across time points, bias in answers cannot be fully excluded. Finally, the present study only examined parents of preschool children that have attended preschool before the start of the pandemic. Future studies are needed to investigate the impact of lockdowns on children that have been supervised at home or via preschool-independent caregivers / educators.

## 5. Conclusion

Overall, the present study has found a rapid increase in internalizing and externalizing problems in preschool children during the second nationwide COVID-19 lockdown in Germany and identified parental stress, parental anxious symptomatology and children's attachment as positively associated factors and parental education and preschool attendance as negatively associated factors. Lockdown measures are necessary means for controlling the spread of the pandemic, nevertheless, their impact on the mental health of parents and preschool children needs to be considered. Children's preschool attendance may be an important aid to alleviate the consequences of lockdown measures by reducing parental burden and children's internalizing problems. Future work should examine the impact of preschool attendance in more detail and determine which benefits of preschool aid in reducing negative consequences.

## Acknowledgments

We would like to thank all parents for their participation and all involved educators for their help in recruiting for the study.

## Author Contributions

**Conceptualization:** Irina Jarvers, Daniel Schleicher, Romuald Brunner, Stephanie Kandsperger.

**Data curation:** Irina Jarvers.

**Formal analysis:** Irina Jarvers.

**Funding acquisition:** Romuald Brunner.

**Investigation:** Irina Jarvers, Angelika Ecker, Daniel Schleicher, Stephanie Kandsperger.

**Methodology:** Irina Jarvers.

**Resources:** Romuald Brunner.

**Software:** Irina Jarvers, Angelika Ecker.

**Supervision:** Romuald Brunner.

**Validation:** Irina Jarvers.

**Writing – original draft:** Irina Jarvers.

**Writing – review & editing:** Irina Jarvers, Angelika Ecker, Daniel Schleicher, Romuald Brunner, Stephanie Kandsperger.

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
