## [Decision Letter · Decision Letter 0]

8 Nov 2022

PONE-D-22-20348Impact of preschool attendance, parental stress, and parental mental health on internalizing and externalizing problems during COVID-19 lockdown measures in preschool childrenPLOS ONE

Dear Dr. Jarvers,

Thank you for submitting your manuscript to PLOS ONE. After careful consideration, we feel that it has merit but does not fully meet PLOS ONE’s publication criteria as it currently stands. Therefore, we invite you to submit a revised version of the manuscript that addresses the points raised during the review process.

We look forward to receiving your revised manuscript.

Kind regards,

Md. Tanvir Hossain

Academic Editor

PLOS ONE

Reviewers' comments:

Reviewer's Responses to Questions

**Comments to the Author**

1. Is the manuscript technically sound, and do the data support the conclusions?

Reviewer #1: Partly

Reviewer #2: Yes

2. Has the statistical analysis been performed appropriately and rigorously? 

Reviewer #1: No

Reviewer #2: Yes

3. Have the authors made all data underlying the findings in their manuscript fully available?

Reviewer #1: Yes

Reviewer #2: No

4. Is the manuscript presented in an intelligible fashion and written in standard English?

Reviewer #1: Yes

Reviewer #2: Yes

5. Review Comments to the Author

Reviewer #1: Introduction

Line no 2 ,“the leading cause of disability” does this mean number one cause of disability worldwide?

Lines 96 – 99; It’s not clear whether you questioned the parents once about all three time points

It is important for an international reader to know what sort of lockdown was in place? Were the preschools completely closed? Were there day care facilities available

Methods

How did the participants access the online tool? Was there any kind of invitations sent for the email lists obtained through preschools? If so, what was the response rate?

This survey is done among a non-random sample. Therefore, the external validity of the study is questionable.

Line 145; Why was three tools used to assess internalizing symptoms

The tools used to measure attachment, parental stress and parental mental health, measures the status at the point of responding and they have been validated for that purpose. Asking about these retrospectively, using tools that are not originally intended for that purpose is quite questionable

Results

Line 331; unless you intend to validate the CBCL, CMFWQ why are you looking at their correlations with SDQ? If you use only SDQ in the main analysis, why were these tools used?

Discussion

Line 496; In my opinion, this is a major flaw in the methodology.

Reviewer #2: Overall, the topic is important and interesting.

I would suggest to tighten the introduction section.

I do not see a conclusion section. I suggest to separate conclusion from discussion.

In conclusion please provide key message found from the study and future research directions.

6. PLOS authors have the option to publish the peer review history of their article (what does this mean?). If published, this will include your full peer review and any attached files.

Reviewer #1: No

Reviewer #2: No

---

## [Author Response · Author response to Decision Letter 0]

21 Nov 2022

We are very grateful for the helpful comments provided by the reviewers. Please find our detailed response in the attached file named 'Response to Reviewers'.

Yours sincerely,

the authors.

---

## [Decision Letter · Decision Letter 1]

30 Jan 2023

Impact of preschool attendance, parental stress, and parental mental health on internalizing and externalizing problems during COVID-19 lockdown measures in preschool children

PONE-D-22-20348R1

Dear Dr. Jarvers,

We’re pleased to inform you that your manuscript has been judged scientifically suitable for publication and will be formally accepted for publication once it meets all outstanding technical requirements.

Kind regards,

Md. Tanvir Hossain

Academic Editor

PLOS ONE

Reviewers' comments:

Reviewer's Responses to Questions

**Comments to the Author**

1. If the authors have adequately addressed your comments raised in a previous round of review and you feel that this manuscript is now acceptable for publication, you may indicate that here to bypass the “Comments to the Author” section, enter your conflict of interest statement in the “Confidential to Editor” section, and submit your "Accept" recommendation.

Reviewer #1: All comments have been addressed

Reviewer #2: All comments have been addressed

2. Is the manuscript technically sound, and do the data support the conclusions?

Reviewer #1: Yes

Reviewer #2: Yes

3. Has the statistical analysis been performed appropriately and rigorously? 

Reviewer #1: Yes

Reviewer #2: Yes

4. Have the authors made all data underlying the findings in their manuscript fully available?

Reviewer #1: Yes

Reviewer #2: Yes

5. Is the manuscript presented in an intelligible fashion and written in standard English?

Reviewer #1: Yes

Reviewer #2: Yes

6. Review Comments to the Author

Reviewer #1: Thanks for attending to all the comments. Although I'm not very convinced about using three tools to assess the same variable, I think this work would add some good evidence to the scientific world.

Reviewer #2: Dear authors,

Your manuscript has been accepted. The comments I raised were successfully addressed.

7. PLOS authors have the option to publish the peer review history of their article (what does this mean?). If published, this will include your full peer review and any attached files.

Reviewer #1: No

Reviewer #2: **Yes: **Professor Dr. Shah Md. Atiqul Haq

---

## [Editor Report · Acceptance letter]

3 Feb 2023

PONE-D-22-20348R1 

Impact of preschool attendance, parental stress, and parental mental health on internalizing and externalizing problems during COVID-19 lockdown measures in preschool children 

Dear Dr. Jarvers:

I'm pleased to inform you that your manuscript has been deemed suitable for publication in PLOS ONE. Congratulations! Your manuscript is now with our production department. 

Kind regards, 

on behalf of

Dr. Md. Tanvir Hossain 

Academic Editor

PLOS ONE